# Hydroxytyrosyl Eicosapentaenoate as a Potential Antioxidant for Omega-3 Fatty Acids: Improved Synthesis and Comparative Evaluation with Other Natural Antioxidants

**DOI:** 10.3390/biom15050714

**Published:** 2025-05-13

**Authors:** Natalia García-Acosta, Rosa Cert, Marta Jordán, Luis Goya, Raquel Mateos, Jose Luis Espartero

**Affiliations:** 1Department of Metabolism and Nutrition (DMN), Institute of Food Science, Food Technology, and Nutrition (ICTAN), Spanish National Research Council (CSIC), C/José Antonio Novais 6, 28040 Madrid, Spain; mjordan@ictan.csic.es (M.J.); luisgoya@ictan.csic.es (L.G.); raquel.mateos@ictan.csic.es (R.M.); 2Fat Institute, Spanish National Research Council (CSIC), Universidad Pablo d Olavide, Edificio 46 Ctra. De Utrera Km 1, 41013 Seville, Spain; rosacert@ig.csic.es; 3CIBER de Diabetes y Enfermedades Metabólicas Asociadas (CIBERDEM), Instituto de Salud Carlos III (ISCIII), 28029 Madrid, Spain; 4Department of Organic and Pharmaceutical Chemistry, Faculty of Pharmacy, University of Seville, 41012 Sevilla, Spain

**Keywords:** hydroxytyrosol, omega-3 fatty acids, antioxidant capacity, enzymatic synthesis, hydroxytyrosyl eicosapentaenoate

## Abstract

Hydroxytyrosol (HT), the primary phenolic compound in virgin olive oil, has notable cardiovascular benefits, particularly in preventing low-density lipoprotein (LDL) oxidation. However, its hydrophilicity limits its solubility and integration into lipid-based formulations. This study aimed to enhance its lipophilicity by synthesizing hydroxytyrosyl eicosapentaenoate (HT-EPA), a derivative of HT and eicosapentaenoic acid (EPA), using a one-step enzymatic catalysis with lipase B from *Candida antarctica* (CALB). The reaction, performed as a suspension of HT in ethyl eicosapentaenoate (Et-EPA) (1:9 molar ratio) under vacuum, achieved higher yields and shorter reaction times than previously reported, with a purity exceeding 98%, confirmed by ^1^H-NMR. For the first time, the antioxidant capacity of HT-EPA in comparison with other natural antioxidants was assessed using the FRAP assay, while its oxidative stability in an omega-3-rich oil matrix was evaluated via the Rancimat method. HT-EPA and hydroxytyrosyl acetate (HT-Ac) displayed antioxidant activity comparable to HT but significantly higher than α-tocopherol, a common food antioxidant. Given the scarcity of effective lipid-soluble antioxidants, HT-EPA represents a promising candidate for omega-3 nutraceuticals, offering enhanced stability and potential health benefits. This study provides a simple, efficient, and scalable strategy for developing functional lipid-based formulations with cardioprotective potential by improving HT solubility while preserving its antioxidant properties.

## 1. Introduction

Cardiovascular diseases (CVDs) remain the leading cause of mortality worldwide, accounting for approximately one-third of all deaths worldwide [1]. Among the many contributing factors, elevated triglycerides (TGs) have emerged as a critical marker of increased cardiovascular risk, particularly in patients with established comorbidities [2,3]. Despite advancements in standard therapies targeting cholesterol and blood pressure, residual cardiovascular risk remains high in many patients [4,5], reinforcing the urgent need for adjunctive interventions capable of effectively mitigating this burden.

One promising avenue for reducing TG levels lies in nutritional interventions, particularly through the use of omega-3 fatty acids (O3FAs), especially eicosapentaenoic acid (EPA) and docosahexaenoic acid (DHA). Among these, EPA is considered more potent in its therapeutic effects [6] and has demonstrated significant lipid-lowering effects in numerous clinical studies [7,8,9,10]. Despite their efficacy, the therapeutic potential of O3FAs is often limited by their inherent instability and susceptibility to oxidation, particularly during formulation and storage [11,12]. Oxidized O3FAs not only exhibit reduced therapeutic value but may also contribute to oxidative stress, thereby counteracting their expected cardiovascular benefits [13].

To address this challenge, incorporating antioxidants as stabilizing agents has gained significant attention [14,15]. Hydroxytyrosol (HT) is a phenolic compound found in virgin olive oil (VOO) and extra virgin olive oil (EVOO), mainly as secoiridoid derivatives (the aldehydic form of oleuropein aglycone and the dialdehydic form of decarboxymethyl oleuropein aglycone). It is also present in smaller amounts as the free form and as the acylated derivative hydroxytyrosol acetate (HT-Ac) [16,17]. HT stands out as a potent antioxidant with well-documented cardiovascular benefits [18,19]. Accumulating evidence from both animal and human studies supports the cardiometabolic benefits of olive-derived phenolic compounds, particularly on lipid profiles and insulin resistance, highlighting their potential as adjuvant agents in the management of metabolic syndrome [20,21,22]. The ability of HT to neutralize oxidative stress suggests a potential role in stabilizing lipid molecules to enhance the efficacy and stability of O3FA-based therapies, apart from its additional bioactivity that could represent a double benefit.

However, HT incorporation into O3FA-based nutraceuticals may present challenges due to its hydrophilic nature, which limits its solubility in lipid-based matrices. One promising strategy to overcome this limitation is the transformation of HT into lipidic derivatives, such as HT esters [23]. This reaction not only enhances the solubility of HT in oily formulations but could also potentiate its antioxidant effects, contributing to the reduction of oxidative stress and improving the lipid profile of the final O3FA products.

This study aimed to optimize the synthesis of HT-EPA (3, see Figure 1), a promising lipophilic antioxidant, for the stabilization of ethyl eicosapentaenoate (Et-EPA) and the development of a new food-grade antioxidant formulation. To this end, this study evaluated both its antioxidant activity and the supplement’s oxidative stability in comparison with HT (1) and another natural lipophilic compound (HT-Ac, 2). Although naturally present in EVOO, HT-Ac was also synthesized in this study for further investigation due to its lipophilic and potential antioxidant characteristics. They were all compared with α-tocopherol, a commonly used antioxidant in the formulation of O3FAs-based nutraceuticals.

## 2. Materials and Methods

### 2.1. Materials

HT and Et-EPA were provided by Indagoo (Cymit Química, Barcelona, Spain). Immobilized *Candida antarctica* lipase B (CALB, Novozym-435^®^) and 2-methyl tetrahydrofuran (MeTHF) were purchased from Sigma-Aldrich (Madrid, Spain), as well as α-tocopherol, (±)-6-hydroxy-2,5,7,8-tetramethylchromane-2-carboxylic acid (Trolox) and 2,4,6-Tris(2-pyridyl)-s-triazine (TPTZ). All other chemicals used as solvents were provided by Cosela (Seville, Spain).

NMR spectra were recorded on a Bruker AVANCE NEO 500 spectrometer operating at 500.13 MHz (^1^H) and 125.75 MHz (^13^C). Samples were dissolved (10 mg/mL) in hexadeuterated dimethyl sulfoxide (DMSO-*d*_6_), and spectra were recorded at 303 K. Chemical shifts are given in parts per million (ppm) with the residual solvent signals (2.49 ppm for ^1^H and 39.5 ppm for ^13^C) as references, whereas coupling constants are given in Hz. 2D-NMR experiments (COSY, HSQC and HMBC) were performed to assign all the signals in the NMR spectra. High-resolution mass spectra were recorded on an Orbitrap Elite (ThermoScientific, Waltham, MA, USA), a hybrid ion trap–orbitrap mass spectrometer with a resolution higher than 240,000, with ESI, HESI, APCI and nanoESI ionization sources.

### 2.2. Synthetic Procedures

#### 2.2.1. Synthesis of Hydroxytyrosyl Acetate (HT-Ac, 2)

To a solution of hydroxytyrosol (4 mmol) in ethyl acetate (AcOEt, 60 mL) was added *p*-toluenesulfonic acid (*p*-TsOH, 200 mg), and the mixture was heated to reflux for 4 h. The progress of the reaction was monitored by thin-layer chromatography (TLC) using a hexane/*tert*-butyl methyl ether (TBME) (1:1) eluent system, until the complete conversion of the starting material to HT-Ac (2). The resulting suspension was washed with a NaHCO_3_-saturated solution, dried with Na_2_SO_4_, and purified by column chromatography to yield pure **2** as a white solid (86% yield). All spectroscopic data were in good accordance with those previously reported [24].

#### 2.2.2. Synthesis of Hydroxytyrosyl Eicosapentaenoate (HT-EPA, 3)

A mixture of HT (**1**, 13 mmol), 5 mL of MeTHF, Et-EPA (120 mmol), and 2 g of Novozyme-435 was stirred under vacuum at 48 °C for 3 h. Then, the resulting suspension was diluted with hexane, filtered through celite to remove the enzyme, and concentrated under reduced pressure. Finally, the oily crude was purified using a flash chromatography column over silica gel using hexane/ethyl acetate (9:1) as eluent. Compound **3** was obtained as a colorless oil in 59% of yield: ^1^H NMR (500 MHz, DMSO-*d*_6_, 303 K), δ (ppm) 8.76 (bs, 1H, OH-6), 8.70 (bs, 1H, OH-5), 6.61 (d, J = 8.0 Hz, 1H, H-7), 6.58 (d, J = 2.1 Hz, 1H, H-4), 6.43 (dd, J = 2.0 and 8.0 Hz, 1H, H-8), 5.32 (m, 10H, H-5’, H-6’, H-8’, H-9’, H-11’, H-12’, H-14’, H-15’, H-17’, and H-18’), 4.09 (t, J = 7.1 Hz, 2H, H-1), 2.79 (t, J = 5.1 Hz, 4H, H-7’ and H-16’), 2.75 (m, 4H, H-10’ and H-13’), 2.66 (t, J = 7.1 Hz, 2H, H-2), 2.24 (t, J = 7.4 Hz, 2H, H-2’), 2.00 (m, 4H, H-4’ and H-19’), 1.54 (q, J = 7.4 Hz, 2H, H-3’), 0.89 (t, J = 7.5 Hz, 3H, H-20’); ^13^C NMR (125 MHz, DMSO-*d*_6_, 303 K), δ (ppm) 172.8 (C-1’), 145.15 (C-5), 143.80 (C-6), 131.65 (C-18’), 129.07 (C-5’), 128.61 (C-15’), 128.46 (C-3), 128.20–127.77 (C-6’, C-8’, C-9’, C-11’, C12’, and C-14’), 127.0 (C-17’), 119.51 (C-8), 116.21 (C-4), 115.55 (C-7), 64.74 (C-1), 33.88 (C-2), 33.00 (C-2’), 26.04 (C-4’), 25.30–25.20 (C-7’, C-10’, C-13’, and C-16’), 24.22 (C-3’), 20.12 (C-19’), and 14.18 (C-20’). HRMS, calc. 461.2662 for C_28_H_38_O_4_Na [M + Na]^+^, found 461.2655 (−1.6 ppm).

### 2.3. Antioxidant Activity Determinations

#### 2.3.1. Ferric Reducing Antioxidant Power (FRAP) Assay

The FRAP assay was carried out according to the procedure described by Benzie and Strain [25], with adaptations for a 96-well microplate format. The antioxidant potential of the synthesized compounds (**2** and **3**), along with HT (**1**) and α-tocopherol, was assessed based on their ability to reduce the TPTZ-Fe (III) complex to its stable ferrous form (TPTZ-Fe (II) complex). Briefly, 60 μL of freshly prepared FRAP reagent, prewarmed at 37 °C, was dispensed into each well of a clear 96-well plate containing 10 μL of the test sample (dissolved in methanol or aqueous-organic mixture), 30 μL of distilled water, and 200 μL of 0.3 M acetate buffer (pH 3.6). The total reaction volume per well was 300 μL, resulting in a final sample dilution of approximately 1:30. The FRAP reagent consisted of 2.5 mL of 10 mmol/L TPTZ in 40 mmol/L HCl, 2.5 mL of 20 mmol/L FeCl₃·6H₂O, and 7.5 mL of acetate buffer. Absorbance at 595 nm was recorded every 90 s for 30 min using a BioTek microplate reader (Gen5 3.11 software), with temperature maintained at 37 °C. Antioxidant capacity was calculated from the absorbance at 30 min using Trolox standard solutions (100–800 μmol/L in methanol), and the FRAP values were expressed as millimolar Trolox equivalent (mM TE). Each value represents the average of three or six determinations.

#### 2.3.2. ABTS Assay

The free radical scavenging capacity was measured according to Re et al.’s (1999) ABTS decoloration method [26], with some modifications for microplate format using a BioTek microplate reader (Gen5 software) equipped for automated dispensing. The method is based on the ability of different components to scavenge the ABTS radical cation (ABTS^+^·) compared to a standard antioxidant (Trolox). The ABTS^+^ radical was generated by reacting 7 mM ABTS with 2.45 mM potassium persulfate in the dark for 12–16 h, and diluted in PBS to an absorbance of 0.70 ± 0.02 at 730 nm. In clear 96-well plates, 10 µL of the sample and 230 µL of solvent were added, followed by 60 µL of the ABTS^+^·solution using the microplate reader. Plates were shaken for 15 s, and absorbance was recorded at 658 nm every 90 s for 30 min at 37 °C. Antioxidant capacity was determined from the area under the curve (AUC_0–30_ min) using Trolox standard curves (50–600 µM). Results were expressed in mM TE (millimolar Trolox equivalent). All analyses were run in triplicate.

#### 2.3.3. Rancimat^®^ Test

The oxidative stability was evaluated using an accelerated automated test with Rancimat equipment (Model 743, Metrohm Co., Basel, Switzerland) subjecting the samples to the action of oxygen at 80 ± 1 °C because at 100 °C they were altered in a very short time (<1 h), which made it impossible to determine the differences between the treatments.

Aliquots of the glyceridic matrix (Et-EPA) were purified according to the ‘free solvent’ procedure. Briefly, 200 g of oil was poured into a glass chromatography column (45 × 3 cm id) packed with 100 g of alumina activated at 200 °C for 3 h. The purified matrices, free of antioxidants, were stored at −18 °C under nitrogen.

An aliquot of 2.5 g of the purified Et-EPA was supplemented with methanolic solutions of hydroxytyrosol esters (**2** and **3**, see Figure 1), HT (**1**), and α-tocopherol at concentrations ranging from 0.5 to 20 mM (the latter selected to achieve a steady state of oxidative stability). 

Then, 0.5 mL of acetone was added to Rancimat vessels, and the mixtures were homogenized. The vessels were then sealed with the measuring heads, placed into the Rancimat apparatus at room temperature, and heated under an air flow rate of 4 L/h. Once the temperature reached 80 °C, the vessel outlets were connected to the conductivity cells, the air flow was increased to 15 L/h, and time measurement was initiated.

The time required to observe a sharp increase in conductivity is defined as the induction time (IT) and is expressed in hours, and represents the stability of the lipid matrix. IT was determined by the intersection of the baseline with the tangent to the conductivity curve. All measurements were performed in duplicate.

### 2.4. Statistical Analysis

FRAP and ABTS data were subjected to a one-way ANOVA to assess differences in antioxidant activity among the tested compounds. Before the analysis, assumptions regarding independence, normality, homogeneity of variances, and the absence of influential observations were evaluated using graphical and statistical methods, with no major violations detected. Post hoc comparisons were conducted using the Tukey–Kramer procedure, which is suitable for unbalanced designs, to explore pairwise differences and group compounds into homogeneous subsets based on mean similarity.

Rancimat statistical analysis was carried out by fitting a two-parameter model, followed by pairwise comparisons of the maximum oxidation rates using Welch’s *t*-test in R Studio (4.4.3 version). Mean differences were considered significant when *p* < 0.05.

## 3. Results

### 3.1. Preparation and Characterization of HT Esters

Hydroxytyrosyl acetate (HT-Ac, **2**) was prepared by a transesterification reaction between HT (**1**) and ethyl acetate under acidic catalytic conditions (Figure 1), as previously described [23].

On the other hand, HT-EPA was synthesized via transesterification of the ethyl ester of EPA (Et-EPA) with HT (**1**), using immobilized CALB (Novozyme-435) as the catalyst (Figure 1), by a modified procedure from that previously reported [27]. To enhance the solubility of HT in the lipophilic Et-EPA, MeTHF, an eco-friendly solvent [28], was initially added, which was rapidly evaporated at the beginning of the reaction. The esterification was carried out under a vacuum to eliminate the ethanol that had formed, which shifts the equilibrium towards the formation of the desired product. A series of experiments were conducted for the optimization of the yield, varying the molar ratios of the substrates (1:23 to 1:9), the amount of catalyst, and reaction time. The optimal conditions were selected based on maximum product yield (59%) and reaction time (3 h). After filtering the enzyme, the excess of Et-EPA was effectively eliminated by flash chromatography, and the structure of compound **3** was confirmed by spectroscopic methods. Its ^1^H NMR spectrum is shown in Figure 1, in comparison with those for HT (**1**) and HT-Ac (**2**).

The figure shows the ^1^H-NMR spectra of HT, HT-Ac, and HT-EPA (**1**,**2**,**3**, respectively), evidencing the structural modifications. HT (**1**) shows characteristic aromatic proton signals between 6.5 and 7.5 ppm, an alcoholic hydroxyl group (-OH) signal around 4.5 ppm, and two broad signals corresponding to phenolic hydroxyl groups in the 8–9 ppm region. In HT-Ac (**2**), some changes in signals allowed confirming acetylation: (i) the disappearance of the alcoholic -OH signal, (ii) a new signal at 2.0 ppm corresponding to the methyl protons of the acetyl group, (iii) the triplet corresponding to the CH_2_ protons in position 1 of HT (spectrum a), shifts 0.6 ppm to downfield in HT-Ac (spectrum b), and (iv), the aromatic region remaining significantly unchanged. HT-EPA (**3**) retains the aromatic signals and incorporates signals from olefinic protons of the EPA chain between 5.0 and 6.5 ppm and new signals between 0.8 and 2.5 ppm corresponding to aliphatic protons of the EPA chain, confirming esterification. These changes, including the 0.6 ppm downfield shift of protons in position 1, as in compound 2, confirm the successful synthesis of the hydroxytyrosyl ester **3**. All these ^1^H signals, as well as those for ^13^C, were unambiguously assigned based on 2D NMR experiments (spectra not shown).

### 3.2. Antioxidant Activity Assays

#### 3.2.1. FRAP Assay

The reducing capacity of HT-EPA (**3**) was determined by using the FRAP assay and compared to its precursor HT (**1**), another natural and lipophilic compound, HT-Ac (**2**), and α-tocopherol, as a reference compound. The results, summarized in Table 1, showed no significant differences in antioxidant activity between HT (**1**) and HT-Ac (**2**). In contrast, HT-EPA (**3**) displayed a statistically significant reduction compared to HT (**1**), although the absolute difference was small. All hydroxytyrosol-based compounds exhibited significantly higher reducing power than the widely used antioxidant in the agro-food industry, α-tocopherol. FRAP values are expressed as millimolar Trolox equivalent antioxidant capacity (TEAC, mM).

#### 3.2.2. ABTS Assay

The radical scavenging activities of the tested compounds showed a similar trend to the reducing activity. In this sense, HT-Ac and HT-EPA exhibited similar antioxidant activity, which was significantly lower than HT activity. Again, compounds **1**–**3** presented significantly higher antioxidant capacity than α-tocopherol.

#### 3.2.3. Rancimat Test

The antioxidant efficacy of the newly synthesized compounds was assessed in a lipid matrix composed of antioxidant-free Et-EPA oil using the accelerated Rancimat method at 80 °C. The IT increased proportionally with antioxidant concentration up to 20 mM, indicating a dose-dependent stabilization effect. Beyond this concentration, no further significant improvements in oxidative stability were observed, suggesting a saturation threshold. Figure 2 illustrates the IT values for purified lipid matrices supplemented with increasing concentrations of HT (**1**), hydroxytyrosyl esters (**2**, **3**), and α-tocopherol. Both ester derivatives exhibited comparable antioxidant activity per millimole, with IT values slightly lower than those observed in matrices containing free HT (**1**) at lower concentrations. However, they demonstrated significantly greater oxidative protection in the lipid matrix compared to α-tocopherol, reinforcing their potential as effective stabilizers for omega-3 formulations.

## 4. Discussion

HT derivatives have been widely studied for their potential application as novel food antioxidants [29,30]. The growing demand for fat-soluble antioxidants in the agro-food industry has led to the development of lipophilic HT esters as a viable alternative to expand the currently limited range of available antioxidants [23,31,32,33]. Currently, the market relies primarily on α-tocopherol, which exhibits only moderate antioxidant activity [34,35,36], and synthetic butylated hydrotoluene (BHT), whose safety profile remains controversial due to the limited number of human studies assessing its possible adverse effects [37,38,39]. In this context, enhancing HT lipophilicity could not only improve its bioavailability in nutraceutical applications [40] but also contribute to the antioxidant stability of other ingredients within the formulation, ultimately extending the product’s overall efficacy.

This ability suggests a potential role in stabilizing O3FA-based nutraceuticals [41,42,43], such as oils rich in EPA, which is particularly interesting because of its cardioprotective properties; however, it is highly prone to oxidation due to its polyunsaturated long-chain structure. Thus, HT esterification with EPA may provide a viable strategy for effectively incorporating HT into a lipid matrix while preventing EPA from rapid oxidation. However, challenges remain regarding the yield in the synthesis of O3FA esters. In this study, we aimed to address these challenges by optimizing the synthesis of HT eicosapentaenoate (HT-EPA) and evaluating its antioxidant capacity and oxidative stability in comparison with HT acetate (HT-Ac) and its precursor, HT.

The enzyme-catalyzed esterification has been demonstrated to be a potentially highly efficient method [44] since it provides a high selectivity with only minor limitations, such as temperature sensitivity. This drawback can be mitigated by immobilizing the enzyme in acrylic resin, although economic factors remain a concern. Bulky omega-3 fatty acids continue to pose a challenge to reaction efficiency, as previously discussed. Under our optimized conditions using immobilized lipase from *Candida antarctica B* as the catalyst, no solvent, a molar ratio of 1:9 (HT to EPA), and a reaction temperature of 48 °C, a conversion yield of 59% was achieved.

The optimized conditions for esterifying Et-EPA showed a marked improvement in yield compared to previously reported EPA esterification methods [27,45]. Although higher yields have been reported in some studies [40,46], those typically involve the use of solvents, which may favor conversion but compromise time efficiency. In contrast, our solvent-free approach not only facilitates reproducibility and reduces purification, as shown in similar processes, such as HT oleate esterification [47,48], but also offers an eco-friendlier alternative. Additionally, our reported yield reflects a stricter metric of efficiency, as it refers to the purified compound rather than pre-purification conversion. It is important to highlight that a shorter reaction time (3 h) was sufficient to achieve a comparable yield. It is also important to clarify that we intentionally avoided the use of free fatty acids, not only to avoid the redundancy of synthesizing EPA from Et-EPA, [49] but also to streamline the purification process.

This strategic choice contributes to a more streamlined and potentially scalable synthesis pathway. The use of immobilized lipases in continuous systems has already been demonstrated as a viable route for industrial scale-up [50,51]. When combined with the high functional performance of HT-EPA, its alignment with natural product trends, and the potential use of low-cost feedstocks [52], this approach suggests a favorable cost-effectiveness balance, especially in the context of high-value applications in food and cosmetic formulations.

According to the results summarized in Table 1, the FRAP analysis did not reveal differences between the antioxidant capacity of hydroxytyrosol (**1**) and its derivatives (**2**,**3**) in practical terms. However, a statistically significant difference was observed when these compounds were compared to α-tocopherol, a lipophilic antioxidant commonly used in the food industry. These findings are consistent with previous FRAP-based comparisons of HT and HT-Ac [53], and they offer new insights about the reducing power of HT-EPA. Similarly, other studies have reported no significant differences between HT and HT-Ac or HT-EPA, as well as other lipophilic esters [46,47], although different antioxidant capacity assays (ABTS and DPPH) were used.

In contrast, the ABTS assay revealed statistically significant differences between HT and its esterified forms, suggesting that esterification may influence certain radical-scavenging mechanisms. Nonetheless, all hydroxytyrosol-based compounds showed markedly higher antioxidant capacity than α-tocopherol, confirming their strong potential as lipid-phase antioxidants. Despite these assay-dependent variations, the overall consistency across various electron and/or hydrogen-transfer-based techniques [54] supports the idea that esterification does not substantially compromise the antioxidant capacity of HT, aligning with similar observations in other phenolic esters [55].

Regarding the Rancimat experiment illustrated in Figure 2, both HT-Ac and HT-EPA esters exhibited similar stabilization effects in our matrix, but greater protection against oxidation when compared with α-tocopherol. This result further supports that esterification does not compromise the antioxidant capacity of HT, which efficiently mitigates oxidation in the EPA matrix, an effect not observed with α-tocopherol. Previous evidence has shown that HT esters display comparable antioxidant performance in an oily matrix [21,32], although ITs were proportionally longer in the olive oil matrix (primarily due to its monounsaturated nature) compared to the EPA polyunsaturated oil matrix. To enhance the resolution of IT differences, the assay temperature was lowered from 100 °C to 80 °C. Nevertheless, this adjustment may not have been enough to fully capture the extent of the oxidative stability differences.

To our knowledge, this is the first study to directly assess the oxidative stability in a purified ethyl-EPA matrix, rather than in a complex fish oil, such as anchovy oil [46], which contains a mixture of different fatty acids. This distinction is especially important given our formulation goal of developing soft-gel capsules containing purified ethyl-EPA for human consumption. By using this simplified matrix, we eliminated confounding interactions from other oil components, allowing for a clear understanding of the intrinsic antioxidant performance of HT-EPA. Furthermore, the concentration range was extended up to 20 mM, well beyond the previously reported maximum of approximately 4 mM and likely allowing for clearer differentiation of the stability between HT and its esters.

Looking ahead, future research could explore the long-term oxidative stability of these compounds under realistic storage conditions to better assess their practical applicability. In addition, in vivo studies are needed to investigate their health-related effects, with particular attention to the bioavailability and metabolic fate of these conjugates, in order to determine whether the esterification affects the absorption and physiological activity of hydroxytyrosol.

## 5. Conclusions

The findings in this study reinforce the potential of HT esterification as a strategy to develop potent, lipophilic antioxidants that overcome the limitations of existing fat-soluble antioxidants in the food industry. By improving nutraceutical stability and leveraging the synergistic effects of EPA and HT, this approach not only offers a more effective solution for managing elevated triglyceride levels but also provides a powerful tool to combat oxidative stress, both of which are key drivers of cardiovascular disease. This could ultimately contribute to preventing cardiovascular events, which broadly impact global health.

## Data Availability

The original contributions presented in this study are included in the article.

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
