# Peer review of "Hydroxytyrosyl Eicosapentaenoate as a Potential Antioxidant for Omega-3 Fatty Acids: Improved Synthesis and Comparative Evaluation with Other Natural Antioxidants"

_biomolecules, 2025, doi:10.3390/biom15050714_

Round 1

Reviewer 1 Report

Comments and Suggestions for Authors

The article entitled "Hydroxytyrosyl Eicosapentaenoate as a Potential Antioxidant for Omega-3 Fatty Acids: Improved Synthesis and Comparative Evaluation with Other Natural Antioxidants" by García-Acosta et al. is a concise and elegant presentation of a promising strategy to improve the stability and efficacy of omega-3 formulations. The authors developed a more efficient synthesis method for hydroxytyrosyl eicosapentaenoate (HT-EPA) through enzymatic catalysis using Candida antarctica lipase B, achieving over 98% purity and improved yield in a shortened reaction time. The synthesized HT-EPA was characterized and compared to hydroxytyrosol (HT), hydroxytyrosyl acetate (HT-Ac), and α-tocopherol using the FRAP assay and Rancimat test. The results demonstrated that HT-EPA maintained antioxidant capacity similar to HT and HT-Ac, and notably outperformed α-tocopherol in oxidative stability within an omega-3-rich matrix. These findings highlight HT-EPA’s potential as a lipophilic antioxidant for nutraceutical applications, especially in the stabilization of polyunsaturated fatty acids like EPA. I left detailed comments and suggestions directly in the PDF file, including recommendations for clarity and minor corrections. Overall, this is a good study that might be published after minor revisions.

Author Response

Dear reviewer, 

Thank you very much for taking the time to review our manuscript and for your valuable feedback. Please find the revised version of the manuscript incorporating the suggested changes. The changes have been shaded in yellow (referee 1), green (referee 2), and pink (referee 3). Some parts have been crossed out but 
not removed, so that the reduction we have made could be easily identified.

Comment 1.  Please discuss the following new meta-analysis regarding the effects of 
supplementation with hydroxytyrosol:

1. https://doi.org/10.1080/10408398.2025.2453090 
2. https://doi.org/10.1016/j.freeradbiomed.2021.06.017 
3. https://doi.org/10.1016/j.repc.2021.05.011

In agreement with your comment, a new sentence was added to the introduction section 
(highlighted in yellow, lines 62-65): “Accumulating evidence from both animal and human 
studies supports the cardiometabolic benefits of olive-derived phenolic compounds, particularly 
on lipid profile and insulin resistance, highlighting their potential as adjuvant agents in the 
management of metabolic syndrome [20,21,22]”.

Comment 2. Use the past tense in “aims” and “evaluates”.

Thank you for the observation. These words have been corrected (highlighted in yellow, lines 75 
and 78).

Comment 3. Why only FRAP assay was conducted? Is it not more frequent to conduct multiple 
antioxidant assays in order to assess TAC of evaluated compounds? I suggest 
running at least one more assay, for example DPPH.

Thank you for your comment. We considered it appropriate to perform an additional experiment 
to further evaluate the antioxidant capacity of our compounds, specifically using the ABTS assay. 
This was done to strengthen the robustness of our findings and to confirm that the antioxidant 
potential extends to other mechanisms of action, such as radical scavenging, as suggested by your 
comment and that of Reviewer 3. This additional experiment has been incorporated into the 
Materials and Methods, Results, and Discussion sections (highlighted in yellow), and the corresponding data have been added to Table 1, along with the FRAP results.

On behalf of the authors, I remain at your disposal for any further clarifications.

Sincerely,

Natalia García Acosta.

Reviewer 2 Report

Comments and Suggestions for Authors

The study explores the synthesis of hydroxytyrosyl eicosapentaenoate (HT-EPA), a derivative of hydroxytyrosol and eicosapentaenoic acid, aiming to enhance lipophilicity and antioxidant potential. The compound was synthesized using a one-step enzymatic process, achieving higher purity and yield than previous methods. The antioxidant capacity was compared with other natural antioxidants, showing promising results for application in omega-3 nutraceuticals. It contributes valuable knowledge to the field of nutraceuticals, especially concerning the stability of omega-3 fatty acids. However, some minor revisions could enhance the clarity and impact of the article.

- While the study is robust on a laboratory scale, the authors should discuss the potential for scaling up the synthesis process of HT-EPA. Addressing economic feasibility. In addition, adding potential industrial applications would provide a more comprehensive view of the research's practical implications.

- Providing a detailed comparison of HT-EPA with other antioxidants in terms of cost-effectiveness and ease of incorporation into commercial products could enhance the relevance of the study for industry applications.

- Although the methods are adequately described, adding more granular details about the experimental setup, such as specific conditions for assays, would improve reproducibility.

-  Authors should conduct and report on long-term stability studies of HT-EPA in omega-3 formulations under various storage conditions. This data would strengthen claims about its effectiveness as a stabilizer.

- Authors should highlight potential areas for future research encouraging further investigation and application of the study findings such as exploring bioavailability and health impacts of HT-EPA in vivo.

The suggested changes would enhance the clarity, application, and impact of this research.

Author Response

Dear reviewer,

Thank you very much for taking the time to review our manuscript and for your valuable feedback. Please find attached the revised version of the manuscript incorporating the suggested changes, with your comments shaded in green color.

Comment 1. While the study is robust on a laboratory scale, the authors should discuss the 
potential for scaling up the synthesis process of HT-EPA. Addressing economic 
feasibility. In addition, adding potential industrial applications would provide a 
more comprehensive view of the research's practical implications.

Comment 2. Providing a detailed comparison of HT-EPA with other antioxidants in terms of 
cost-effectiveness and ease of incorporation into commercial products could 
enhance the relevance of the study for industry applications.

We sincerely thank you for these two insightful comments (referring to comments 1 and 2). 
Although the primary objective of our study was to characterize HT-EPA at the laboratory scale, 
we agree that addressing the potential for scale-up and industrial application enhances the broader 
relevance of the work. Accordingly, we have added a new paragraph to the Discussion section 
(highlighted in green, lines 328–334) that briefly addresses these aspects.
Specifically, we note that the enzymatic synthesis approach used in this study has previously been 
shown to be scalable, particularly through the use of immobilized lipases in continuous systems 
[1,2]. While scale-up remains a technical and economic challenge, sourcing hydroxytyrosol from 
agri-food byproducts could significantly reduce raw material costs and improve overall process 
feasibility [3].
In terms of cost-effectiveness and application potential, HT-EPA may offer advantages over 
conventional antioxidants such as BHT or α-tocopherol. These include higher oxidative stability 
in lipid-rich systems [4] and an amphiphilic structure that facilitates its incorporation into food 
and cosmetic formulations [5,6,7,8]. While the synthetic route may be more elaborate than that 
of some synthetic antioxidants, the superior functionality suggests a likely favorable costbenefit balance, particularly for high-value formulations. Although a detailed economic analysis 
is beyond the scope of this study, we believe this discussion provides a useful perspective on the 
compound’s practical potential, in line with your suggestion.

Comment 3. Although the methods are adequately described, adding more granular details 
about the experimental setup, such as specific conditions for assays, would improve 
reproducibility.

According to your comment, we decided to rewrite the FRAP and Rancimat method sections in 
order to improve clarity and reproducibility (highlighted in green, lines 140-148 and 169-187).

Comment 4. Authors should conduct and report on long-term stability studies of HT-EPA in 
omega-3 formulations under various storage conditions. This data would strengthen 
claims about its effectiveness as a stabilizer.

Comment 5. Authors should highlight potential areas for future research encouraging further 
investigation and application of the study findings such as exploring bioavailability 
and health impacts of HT-EPA in vivo.

Thank you for the valuable comments regarding the future directions of our research. Suggestions 
4 and 5 have been addressed by adding a concluding paragraph to the discussion section (lines 
372–377).
In this regard, a human bioavailability study has been conducted to assess hydroxytyrosol 
absorption and metabolism when consumed in the form of these food supplements, aiming to 
compare how esterification may influence its bioavailability. We are currently analysing the 
results, which will be published soon. In parallel, a randomized, double-blind, crossover, chronic 
clinical trial is underway in both healthy individuals and subjects at cardiovascular risk to evaluate 
the benefits of consuming these products over 16 weeks on cardiovascular health markers. The 
intervention is expected to conclude by the end of 2025. At the same time, the long-term stability 
of these supplements is also being investigated.

References: 

[1] Villeneuve, P.; Muderhwa, J.M.; Graille, J.; Haas, M.J. Customizing lipases for biocatalysis: 
A survey of chemical, physical and molecular biological approaches. J. Mol. Catal. B Enzym. 
2000, 9, 113–148.
[2] Fjerbaek, L.; Christensen, K.V.; Norddahl, B. A review of the current state of biodiesel 
production using enzymatic transesterification. Biotechnol. Bioeng. 2009, 102, 1298–1315.
[3] Salta, F.N.; Kosma, I.S.; Lordan, R.; Vlachogianni, I.C.; Ladavos, A.K.; Zampounis, A. Olive 
phenolic compounds: A novel class of nutraceuticals and their impact on human health. Nutrients 
2019, 11, 1325.
[4] Medina, I.; Mateos, R.; Andlauer, W.; Torres de Pinedo, A.; Bravo, L. Stability of bioactive 
compounds in foods: During processing and storage. Crit. Rev. Food Sci. Nutr. 2006, 47, 485–
509.
[5] Barrajón-Catalán, E.; Fernández-Arroyo, S.; Roldán, C.; Guillén, E.; Saura, D.; SeguraCarretero, A.; Micol, V. A systematic study of the antiproliferative activity of polyphenols from 
olive tree against cancer cells. Food Chem. 2011, 132, 493–498.
[6] Roux, S.; Villeneuve, P.; Visioli, F. Amphiphilic polyphenols: A promising class of functional 
ingredients for the food and nutraceutical industry. Trends Food Sci. Technol. 2018, 79, 170–183. 
[7] Annunziata, G.; Maisto, M.; Schisano, C.; Ciampaglia, R.; Narciso, V.; Tenore, G.C.; 
Novellino, E. Resveratrol as a novel anti-aging molecule: Promises and controversies. Phytother. 
Res. 2020, 34 (5), 1146–1155
[8] Choi, J.Y.; Kim, D.W.; Yoon, H.S.; Lee, J.Y.; Lee, S.E.; Shin, S.H.; Lee, J.H. Antiinflammatory and skin-protective effects of hydroxytyrosol, a polyphenol from olive oil, on 
human keratinocytes. Mol. Cell. Toxicol. 2018, 14, 377–384.

On behalf of the authors, I remain at your disposal for any further clarifications.

Sincerely,

Natalia García Acosta.

Reviewer 3 Report

Comments and Suggestions for Authors

Abstract:

  1. All abbreviations should be explained – please refer to the Line 19. 

Introduction:

  1. Lines 58 – 62: Please explain if these two mentioned forms are naturally present in VOO and EVOO, or are chemically synthesized.
  2. The main hypothesis have to be clearly stated.

Material and methods:

  1. Lines 91-82: What was the reason for using CALB in the synthesis procedure? Could enzymes of other origins be used?
  2. Lines 133-134: Why the number of replicates were not equal?
  3. Why Authors focus on the FRAP and Rancimat methods? Please explain if DPPH, ABTS and ORAC may be also used in your study.

Results:

  1. Figure 2 should be prepared in color or the designation of each point should be bigger as in present form it is hard to distinguish.

Discussion

  1. Line 260: ‘oils rich in EPA’ or ‘oils being a good source of EPA’.
  2. Could a similar synthesis method be successfully used to stabilize the antioxidant properties of fats and oils that are sources of conjugated fatty acids (CLA and CLnA), e.g milk fat and pomegranate seed oil? Please explain in the context of extending shelf life.

Author Response

Dear reviewer,

Thank you very much for taking the time to review our manuscript and for your valuable feedback. Please find attached the revised version of the manuscript incorporating the suggested changes. The changes have been shaded in yellow (referee 1), green (referee 2), and pink (referee 3).

Comment 1. All abbreviations should be explained – please refer to the Line 19.

Thank you for the observation. “LDL” has been explained in line 19, as well as in the 
abbreviations section.

Comment 2. Lines 58 – 62: Please explain if these two mentioned forms are naturally present in 
VOO and EVOO, or are chemically synthesized. The main hypothesis has to be 
clearly stated.

Thank you for your valuable comment. In response, we have revised the introduction to better 
distinguish between compounds naturally present in olive oil and those selected for synthesis. 
Specifically, we clarify that hydroxytyrosol (HT) and its acylated derivative, hydroxytyrosol 
acetate (HT-Ac), are naturally present in both virgin and extra virgin olive oil (VOO and EVOO, 
respectively). HT is mainly present as secoiridoid derivatives, although it also occurs in smaller 
amounts in its free form. The natural origin of these compounds is now detailed in lines 57-61. 
At the same time, the rationale for the synthetic work is explained, emphasizing that although 
HT-Ac naturally occurs in VOO and EVOO, it was also synthesized for comparative and 
functional purposes, due to its lipophilic character and potential antioxidant activity (highlighted 
in pink, lines 80–82).
Additionally, we have reformulated the study's objective (with changes highlighted in pink, lines 
75-77) to more accurately reflect our focus on optimizing the synthesis of HT-EPA and evaluating
its functional performance as a lipophilic antioxidant, in comparison with structurally related 
compounds and a commercial standard (α-tocopherol). 

Comment 3. Lines 91-82: What was the reason for using CALB in the synthesis procedure? 
Could enzymes of other origins be used?

Thank you for raising this important point. The selection of Candida antarctica lipase B (CALB) 
was based on its broad applicability in transesterification reactions involving sensitive phenolic 
compounds, due to its robustness, selectivity, and compatibility with mild, non-degradative 
conditions. Its commercial immobilized form (e.g., Novozym® 435) is particularly effective in 
organic media. This choice was further supported by unpublished comparative assays performed 
by Prof. Dr. Espartero’s group, which demonstrated that CALB consistently outperformed other 
commercial lipases under the experimental conditions applied in this study, and by numerous 
published studies [9-12].

Comment 4. Lines 133-134: Why the number of replicates were not equal?

Thank you for this insightful comment regarding the unequal number of replicates across the 
antioxidant compounds tested. We fully understand the relevance of this observation and 
appreciate the opportunity to clarify our rationale. 
HT and α-tocopherol served primarily as reference compounds, whose antioxidant behavior is 
well established and highly consistent, both in the literature and in our preliminary assays. This 
justified a smaller number of replicates without compromising statistical validity. In contrast, the 
ester derivatives constituted the main focus of our investigation and were expected to show greater 
variability due to their structural novelty and differences in physicochemical properties. 
Increasing the number of replicates allowed us to improve statistical power and obtain more 
robust effect estimates.
Additionally, the number of HT replicates was constrained by limited sample availability at the 
time of testing. It is also worth noting that the study was conducted in two stages, with HT 
assessed first as a baseline, and additional replicates for the esters being incorporated in the second 
phase based on the initial results.

Comment 5. Why Authors focus on the FRAP and Rancimat methods? Please explain if DPPH, 
ABTS and ORAC may be also used in your study.

Thank you for your comment. We considered it appropriate to perform an additional experiment 
to further evaluate the antioxidant capacity of our compounds, specifically using the ABTS assay. 
This was done to strengthen the robustness of our findings and to confirm that the antioxidant 
potential extends to other mechanisms of action, such as radical scavenging, as suggested by your 
comment and that of Reviewer 1. This additional experiment has been incorporated into the 
Materials and Methods, Results, and Discussion sections (highlighted in yellow), and the 
corresponding data have been added to Table 1, along with the FRAP results.

Comment 6. Figure 2 should be prepared in color or the designation of each point should be 
bigger as in present form it is hard to distinguish.

We sincerely appreciate this observation. In response, Figure 2 has been revised by incorporating 
color to enhance the clarity and readability of the results.

Comment 7. Line 260: ‘oils rich in EPA’ or ‘oils being a good source of EPA’.

Thank you for the suggestion. It has been revised in the text to ensure better understanding 
(highlighted in pink, line 310).

Comment 8. Could a similar synthesis method be successfully used to stabilize the antioxidant 
properties of fats and oils that are sources of conjugated fatty acids (CLA and 
CLnA), e.g milk fat and pomegranate seed oil? Please explain in the context of 
extending shelf life.

Thank you for this interesting question. We strongly believe that a similar synthetic strategy 
could be applied to fats and oils rich in conjugated fatty acids, including milk fat and 
pomegranate seed oil. Given that conjugated fatty acids are highly susceptible to oxidation 
due to their multiple double bonds, stabilizing them through esterification with a phenolic 
compound possessing strong antioxidant properties appears to be a promising approach.
Hydroxytyrosol has demonstrated excellent radical scavenging capacity and lipid protection 
in various systems, and its covalent incorporation could help extend the oxidative stability 
and shelf life of these oils. Since the chemical structure and reactivity of CLA and CLnA are 
compatible with transesterification conditions, we believe this strategy could be adapted with 
minimal modifications to achieve similar protective effects.

References:

[9] Gotor-Fernández, V.; Busto, E.; Gotor, V. Candida antarctica Lipase B: An Ideal Biocatalyst 
for the Preparation of Nitrogenated Organic Compounds. Adv. Synth. Catal. 2006, 348, 797–812. 
[10] Sharma, R.; Chisti, Y.; Banerjee, U.C. Production, purification, characterization, and 
applications of lipases. Biotechnol. Adv. 2001, 19, 627–662. 
[11] Zoumpanioti, M.; Merianou, E.; Karandreas, T.; Stamatis, H.; Xenakis, A. Esterification of 
phenolic acids catalyzed by lipases immobilized in organogels. Biotechnol. Lett. 2010, 32, 1457–
1462. 
[12] BiaÅ‚ecka-FlorjaÅ„czyk, E.; Fabiszewska, A.; Zieniuk, B. Phenolic Acids Derivatives –
Biotechnological Methods of Synthesis and Bioactivity. Curr. Pharm. Biotechnol. 2018, 19, 
1098–1113

On behalf of the authors, I remain at your disposal for any further clarifications.

Sincerely,
